# Age-Induced Differential Changes in the Central and Colonic Human Circadian Oscillators

**DOI:** 10.3390/ijms21020674

**Published:** 2020-01-20

**Authors:** Cristina Camello-Almaraz, Francisco E. Martin-Cano, Francisco J. Santos, Mª Teresa Espin, Juan Antonio Madrid, Maria J. Pozo, Pedro J. Camello

**Affiliations:** 1Department of Physiology, Institute of Molecular Pathology Biomarkers, University of Extremadura, Campus Universitario, 10003 Cáceres, Spain; mcca@unex.es (C.C.-A.); fmartincano@gmail.com (F.E.M.-C.); mjpozo@unex.es (M.J.P.); 2Surgery Department, University Hospital, Servicio Extremeño de Salud, Avda Universidad, 10004 Cáceres, Spain; cirugiasantos@gmail.com; 3Faculty of Medicine, Infanta Cristina University Hospital, Servicio Extremeño de Salud, Avda Elbas, 06080 Badajoz, Spain; mariateresa.espin@gmail.com; 4Chronobiology Lab, Department of Physiology, College of Biology, University of Murcia, Mare Nostrum Campus, IMIB-Arrixaca, 30100 Murcia, Spain; jamadrid@um.es

**Keywords:** circadian rhythms, colon, smooth muscle, clock genes, contractility, calcium signals

## Abstract

Aging modifies not only multiple cellular and homeostatic systems, but also biological rhythms. The circadian system is driven by a central hypothalamic oscillator which entrains peripheral oscillators, in both cases underlain by circadian genes. Our aim was to characterize the effect of aging in the circadian expression of clock genes in the human colon. Ambulatory recordings of the circadian rhythms of skin wrist temperature, motor activity and the integrated variable TAP (temperature, activity and position) were dampened by aging, especially beyond 74 years of age. On the contrary, quantitative analysis of genes expression in the muscle layer of colonic explants during 24 h revealed that the circadian expression of *Bmal1*, *Per1* and *Clock* genes, was larger beyond that age. In vitro experiments showed that aging induced a parallel increase in the myogenic contractility of the circular colonic muscle. This effect was not accompanied by enhancement of Ca^2+^ signals. In conclusion, we describe here for the first time the presence of a molecular oscillator in the human colon. Aging has a differential effect on the systemic circadian rhythms, that are impaired by aging, and the colonic oscillator, that is strengthened in parallel with the myogenic contractility.

## 1. Introduction

Aging is a complex and multifactorial process which involves a large number of structural and functional alterations from the cellular to the homeostatic levels [1]. The consequence is a reduction in the functional reserve of the organism, ultimately leading to pathological alterations and death. The alterations induced by aging also affect the physiological mechanisms responsible for regulation of body systems and their adaptation to external conditions, including the circadian system [2].

Circadian rhythms are a key factor to maintain homeostasis in vertebrates, adjusting the endocrine and neural regulatory mechanisms to environmental rhythms. The circadian system is composed by a central pacemaker located at the suprachiasmatic nuclei (SCN) of the hypothalamus and the oscillators present at peripheral tissues. Environmental cues such as light-dark cycle entrain the SCN, which operates as central pacemaker and sets in phase the peripheral oscillators [3]. Both the SCN and the peripheral tissues express rhythmically circadian clock genes, which generate circadian oscillations controlling regulatory proteins [4].

The circadian rhythms play a role in the physiological functions of organs and tissues. Gastrointestinal function is known to display rhythms [5], and in the rodent colon a local oscillator, based on clock genes, has been proposed to entrain the motor activity to central rhythms [6,7,8].

Aging is known to impair the circadian system [2,9]. Some studies have reported a disorganization of central circadian rhythms such as body temperature or sleep [10,11,12]. By the contrary, information about the effect of aging on clock genes is relatively scarce and non-conclusive: both impairment and change have been reported for clock gene expression in SCN of animal models [13], while there is no information in the human species.

The objective of the present study was to investigate the effects of aging in the expression of clock genes of the muscular layer of the human colon and the relationship with circadian marker rhythms in normal living conditions.

## 2. Results

To study in normal living conditions the circadian status of the subjects involved in the study we recorded during one week wrist skin temperature, body position, activity and light (see Methods section) with an ambulatory device. In addition, we calculated the variable TAP, which integrates temperature, activity and position to evaluate the circadian status, and the relative amplitude (RA) and circadian functional index (CFI) as quantitative circadian parameters [14].

Figure 1 shows two representative examples of a 7 days average of continuous records of wrist temperature and motor activity, and the calculated variable TAP. As can be observed, both motor activity and TAP values remain at high values during the daytime. By the contrary, wrist temperature shows an opposite behavior, dropping after waking time, while it remains at highest level during the night. This two-levels switching behavior is a known feature of some human circadian rhythms and also in clock genes of gastrointestinal tract [15], so that for quantification the non-parametric circadian estimators CFI and RA were calculated instead of other classic sinusoidal analysis (e.g, cosinor test). We employed RA in the rest of our study because there is a high correlation between both indexes (CFI is calculated with RA and other parameters of the circadian variables) and because CFI was not available in our study of the clock genes expression.

Simple visual examination suggested that aging was associated to weaker rhythms, as judged from the smaller amplitude in aged individuals. To further evaluate this, we studied the correlation between age and the circadian functionality estimator RA for TAP and the underlying variables. Figure 2A shows that age and RA showed a statistically significant correlation for TAP, position and activity (*p* < 0.05 or smaller for Pearson r coefficient), and for temperature (see table insert of Figure 2). 

The coefficient of determination R^2^ also showed that age could explain up to 50% of the variability for these parameters. Detailed inspection of the correlation graphs suggested that circadian function is especially impaired for ages above 70, limiting the fitting of a linear regression model. To investigate this issue we used a segmented regression analysis of the RA/age data, based on minimization of the overall sum of squares of the model by fitting data to two linear segments separated by a break point (see Methods). Both TAP, activity and position showed a statistically improved fitting with a breakpoint at the age of 74, as shown by the table of Figure 2 (right column). After identification of this break point it was possible to compare RA before and after this age. Panel B of Figure 2 reveals that all the circadian variables decrease at this age (*p* < 0.05 or smaller). The CFI index also showed a similar behaviour, with a break point at age 74 and smaller RA for older individuals (not shown). 

Given the role of clock genes in the peripheral circadian rhythms, we studied the circadian expression of these genes in the muscle layer of colonic explants of individuals of different ages. The explants were prepared the same day of the sample collection and sampling for expression determination was initiated at 8:00 am of the next day. Figure 3 shows the expression of *Bmal1*, *Clock* and *Per1* averaged for all the analysed samples. The average data indicate, as expected, a cycle close to 24 h, with a simultaneous increase in *Bmal1* and *Clock* expression, while *PER1* maximum was delayed around 6 h respect to the former. We analysed the average data with the cosinor periodogram, which revealed statistically significant rhythms for the three genes (*Clock* and *Per1 p* < 0.001, *Bmal1 p* < 0.01) Assuming a period close to 24 h (the duration of the sampling protocol) simple cosinor analysis indicated acrophases at 18.7 and 19 h for *Clock* and *Bmal1* and at 24.1 h for *Per1*.

To investigate the correlation between age and the circadian status of gene expression, we calculated the RA of the rhythm of each gene for all the individuals. Plots for RA vs age showed that the rhythm amplitude for the three genes increased with age, as shown in Figure 4. The correlation was statistically significant for the three genes, although in the case of *Per1* the correlation was in the border of significance. This pattern in the relationship with age was therefore opposite to that observed for the main circadian markers described above (Figure 2). This differential effect is better visualized in Figure 4D; direct comparison between the age dependency of TAP and *Clock* rhythms showed a clearly opposite effect of aging in the systemic rhythms (impairment) versus the colonic rhythms (enhancement). 

Although visual inspection of the correlation between RA of genes rhythms vs age did not show a break point we examined separately the rhythms for individuals above and below 74 years of age. The result was a clear enhancement of the rhythms amplitude for the oldest subjects, as can be observed in Figure 5. In fact, cosinor periodogram revealed that below 74 years the rhythm was not statistically significant neither for *Clock* and *Per1* (*p* > 0.17 and 0.37 respectively) although above 74 years it was highly significant for the three genes (*p* < 0.0001). In the case of *Bmal1* the rhythm was significant for samples below 74 years although the significance was lower (*p* < 0.02).

The next step in our study was to investigate whether there is some kind of association between the circadian changes linked to age and the main function of the colonic muscle, contraction. Due to the limited size of the surgical samples used in our study, we were unable to perform contractility studies in most of the samples assayed for gene expression. Therefore, we analysed the contractility in samples from additional subjects (see Methods section) to study the possible changes associated to age. The contractions evoked by 100 µM bethanechol, a depolarizing solution (KCl 60 mM) or 10 µM substance P were enhanced in samples from donors above 74 years of age compared to younger donors, resembling the pattern observed for the systemic circadian rhythms described in Figure 2. This effect was statistically significant for bethanechol and for substance P, as can be seen in Figure 6.

On the other hand, the enhancing effect of age on this myogenic contraction was also accompanied by a slight but statistically significant positive correlation between age and contraction for bethanechol and substance P, with correlation coefficients above 0.5 (Figure 6, right panel). This pattern resembled the positive correlation between age and the relative amplitude of the clock genes rhythms described in Figure 4.

Since cytosolic Ca^2+^ concentration ([Ca^2+^]_i_) is the main determinant of the smooth muscle contraction, we studied whether the contractility changes shown above were associated to modifications in the Ca^2+^ signals of single muscle cells isolated from samples analysed for genes expression. Cells were challenged with three different agonists, the neurotransmitter ACh, depolarization of plasma membrane (KCl-rich solution) and caffeine, a known releaser of Ca^2+^ from intracellular stores in smooth muscle cells. As can be seen in Figure 7A, there is no correlation between *Clock* or *Per1* expression and ACh signal, although a negative correlation was observed in the case of *Bmal1* expression. Correlation was also absent for the responses to caffeine (*Bmal1* R2 = 0.002, *p* < 0.91; *Clock* R2 = 0.197, *p* < 0.27 and *Per1* R2 = 0.001, *p* < 0.99) and to KCl (*Bmal1* R2 = 0.001, *p* < 0.973, *Clock* R2 = 0.011, *p* < 0.805 and *Per1* R2 = 0.222, *p* < 0.238). In addition, we found no correlation between the Ca^2+^ response to any of the three stimuli and age in the individuals analysed for gene expression, as can be seen in Figure 7B. Moreover, determination of [Ca^2+^]_i_ responses performed in samples from other 20 individuals (see Methods section) revealed no significant increases associated to age (by the contrary, a slight decrease above 74 years of age was observed). Correlation in this larger sample was not significant for any of the three studied stimuli (ACh R2 0.002, *p* < 0.821; Caffeine R2 0.028, *p* < 0.373; KCl R2 0.005, *p* < 0.760).

## 3. Discussion

Our study reports for the first time clock genes rhythms in the human colon and reveals a differential effect of aging on the robustness of systemic and colonic circadian rhythms.

Chronodisruption, or impairment and disorganization of circadian rhythms, is a previously reported feature of aging [11,12]. In keeping with it, our study indicates that the marker rhythms (wrist temperature, motor activity and body position), controlled in part by the central pacemaker, lose robustness with age, in spite of the small size of our sample. Also, as expected, we find here that the integrated circadian variable TAP is a useful indicator to study the human circadian behavior in aged individuals, confirming the recent demonstration of its high efficiency to discriminate aged individuals [16]. 

To our knowledge this is the first report of clock genes rhythms in human colon, because previous reports in this organ only communicate single time measurements of clock gene expression [17,18]. In addition, information on the circadian rhythms of clock genes in aged humans is scarce. Previous reports found no change with age in hair tissue [19] nor in tendon [20]. Moreover, age-dependent reductions have been described in the amplitude of *Per1*/2 genes rhythms of human brain [21] and in the clock genes expression of human granulosa ovarian cells [22]. We describe here an increase in the amplitude of human peripheral clock genes, a novel finding to our knowledge. A previous study in mouse reported no effects of aging in the expression of clock genes in colon and small intestine [23], in line with most of evidences from other tissues [13].

In the classical regulatory model of circadian oscillations *Per1* expression is expected to peak in antiphase to *Bmal1* and *Clock*. However, in this study the shift in phase is less than 6 h, as can be seen in Figure 3. Other authors have shown similar shifts in human molecular rhythms, as is the case for the 3–7 h shift in leukocytes [24] or for the absence of shift in visceral fat [25]. Whether this is due to variability in human studies or to the presence of additional regulatory mechanisms of the molecular clock is out of the scope of our work.

A limitation of our study is that the feeding schedule of the subjects was not controlled. Given that meal time has been shown to synchronize epithelial proliferation in the mice colon [26], it could be argued that the effects of aging on the colon rhythms are due to differences in the feeding pattern of elderly individuals. However, to be certain it would be necessary that aging produces the same modification of the meal pattern in all the individuals in free living conditions, which is an unlikely condition. In addition, to influence the genes expression in the explants, the effect of the meal pattern should endure more than 24 h due to the preoperative fasting. The same line of argument makes unlikely that the change in the colon clock is due to impairment of the entraining role of the SCN; in vitro the explants oscillate free of the regulatory mechanisms.

The variability of effects of aging on the peripheral clocks could be due not to experimental issues but to a genuine differential effect of age. Thus, Wyse and Coogan [27] found different effects in several brain locations in mouse. The underlying reason could be that aging induces not a widespread loss of peripheral clock genes, but an impairment of the entraining control from the central oscillator, as proposed previously [12]. We have shown in mice that the colon content of melatonin is enhanced in aged mice, an effect suppressed by melatonin treatment [28]. Aging also alters glucocorticoid levels, which have been reported to entrain gut clock genes [29]. This would be also in keeping with the finding that peripheral oscillators are less sensitive than the central oscillator to changes in external time-setting stimuli (reviewed in [13]), although chronodisruption induced by external conditions alters the molecular clocks of colon and other gastrointestinal organs in rat [30,31].

The alterations induced by age in the molecular clock of the human colon could have clear functional consequences. The gastrointestinal tract displays circadian rhythms for several functions, from cell proliferation to motility, and also plays a role as a circadian regulator through release of melatonin from the neuroendocrine cells of the intestinal wall [5]. A detailed series of reports has demonstrated in the mouse colon a circadian molecular clock in the myenteric plexus and the epithelial layer that could impact on motility [7,8]. The human colon presents also circadian changes in terms of mass movements and frequency of bowel symptoms (diarrhoea, constipation) under circadian disruption [6], a condition also leading to increased incidence of irritable bowel syndrome [32]. 

The circadian disruption discussed above is likely related to the functional changes induced by aging. This condition has a direct deleterious effect in the function of visceral smooth muscle both in animal models [33,34] and in the human gastrointestinal system [35,36]. The fact that melatonin prevents these modifications in visceral smooth muscle [28,34,37] suggests a role of chronodisruption in these changes. The present study shows a significant alteration in the myogenic contractility of the circular colonic muscle, the main component responsible for the segmentation contractions of the colon. It is noteworthy that the changes in the contractility follow the same pattern displayed by the amplitude of the molecular clock, i.e., an enhancement in elderly individuals. Given that our results have been obtained in descending and sigma colon, a zone of resistance to colonic content that delivers it only upon defecation [38], a higher reactivity to agonist could facilitate constipation, which is more frequent in elderly [35,36].

Regarding the link between the changes in the colonic molecular clock and the cellular function, previous evidences indicate that clock genes could control expression of voltage operated Ca^2+^ channels [39] and large conductance Ca^2+^-activates K^+^ channel [40] in brain of animal models, as well as Na^+^/H^+^ [41] and NaCl transporters [42] in rat colon. However, our experimental observations rule out [Ca^2+^]_i_ signals as the downstream mechanism modified by clock genes in aged muscle to enhance the contractile response. Instead of an enhanced [Ca^2+^]_i_ response to the main neurotransmitter, we found a significant negative correlation with clock gene and non-significant decays for three different [Ca^2+^]_i_ mobilizers. This means that the target is another component of the contractile pathway, such as the contractile proteins or the Ca^2+^ sensitizing pathways.

## 4. Materials and Methods 

### 4.1. Subjects

Data were obtained from 12 subjects of ages between 33 and 89, all of them with partial colectomy prescription and free of neurological and metabolic diseases. All subjects gave their informed consent for inclusion before they participated in the study. The study was conducted in accordance with the Declaration of Helsinki, and the protocol was approved by the Ethical Committee of the University of Extremadura (26 January 2011) and the Ethic Committee of Clinic Research of the Extremadura Health Agency (Servicio Extremeño de Salud) at Cáceres (27 January 2011) (“Envejecimiento prematuro del músculo liso: relación con la expresión de genes reloj”, 23/2011).

Some data (contractility and Ca^2+^ signals) were also obtained from 20 additional patients (age range 46–86) part of a research project on the effects of aging on the physiology of colonic muscle. All of them followed the same ethical procedures mentioned above.

### 4.2. Aacquisition and Processing of Circadian Marker Rhythms

Collection of marker rhythms were performed during 7 days and finished at least one week before surgery. Distal skin temperature and light exposure were recorded with a wrist sensor provided with a thermometer and a luxometer, and activity and body position were recorded with an accelerometer and an inclinometer embedded in a bracelet. Sleep time was evaluated with the wrist sensor and a written questionnaire. Time series analysis were performed at the Chronobiology Laboratory of the University of Murcia to obtain the circadian parameters following the method developed by [14]. Briefly, for each individual position (P), activity (A) and temperature (T) variables were normalized (using 95th and 5th percentiles as max and min). Next, since wrist temperature is higher during sleep and lower when the subject is awake (opposite to P and A variables), the normalized T was inverted so that the maximum values for the 3 variables occurred at the same time of the day. The integrating variable TAP was then calculated by averaging the three normalized variables. A value close to 0 indicate rest and sleep, with low activity, horizontal position and high wrist skin temperature, and values near 1 correspond to higher activity and lower temperature. The estimators for circadian robustness relative amplitude (RA) and circadian functional index (CFI) were obtained for all the circadian variables. RA was calculated as (Max − min)/(Max + min). CFI was calculated as an average of three parameters: RA, interdaily stability (0 for Gaussian noise, 1 for exact rhythm repetition) and intradaily variability (0 for Gaussian noise, 1 for perfect sinusoid). RA was calculated also for clock genes expression.

### 4.3. Clock Genes Expression

Sterile samples were obtained from the healthy border of surgical colon resection, placed in physiological solution at 4 °C and processed within 2 h after surgery (at 10–14 h). In the laboratory the sample was placed in physiological Krebs-Henseleit solution (K-HS; mM composition 113 NaCl, 4,7 KCl, 2,5 CaCl_2_, 1,2 KH_2_PO_4_, 1,2 MgSO_4_, 25 NaHCO_3_ and 11,5 D-glucose, pH 7.35 equilibrated with 95% O_2_–5% CO_2_) and extensively washed. Sample was dissected to remove blood vessels, fatty tissue and the mucosal layer, and six pieces of 0.5–1 cm^2^ were placed in a 6 well culture plate in DMEM medium (containing 20% bovine fetal serum, supplemented with penicillin, streptomycin and gentamycin) at 37 °C, 5% CO2. Samples were removed from the incubator every 4 h starting 8 a.m. of the next day, placed in a sterile tube with RNA Later before store at −80 °C until determination of genes expression.

After RNA extraction with MiniRNA isolation Kit (GE Healthcare, Barcelona, Spain) and DNA transcription (inverse transcriptase iTAQ, BioRad, Madrid, Spain) expression of *Clock*, *Per1* and *Bmal1* genes (and 18S as constitutive gene for reference) was determined by RT-PCR using Taqman probes (Applied Biosystems, Foster City, CA, USA) with following references: *Clock* (Hs00231857_m1), *Per1* (Hs00242988_m1), *Bmal1* (Hs0015147_m1) and 18S (Hs99999901_s1). Expression levels were normalized to 18S levels following the 2^−∆∆CT^ method [43]. Cosinor analysis was used to rule out circadian changes in 18S expression.

### 4.4. Contraction Recording of Colonic Smooth Muscle Strips

Once the sample was cut open longitudinally and vessels, fat and the mucosal layer were removed by careful dissection, circularly orientated strips (~5 mm × 10 mm) were cut and placed vertically in a 5-mL organ bath filled with K-HS maintained at 37 °C and gassed with 95% O_2_–5% CO_2_. Isometric contractions were measured using force displacement transducers, digitized using a MacLab hardware unit and dedicated software (ADInstruments, Colorado Springs, CO, USA). An initial 0.5 g resting working load was used and the strips were allowed to equilibrate for 60 min, with solution changes every 20 min. To induce myogenic contraction, strips were challenged with three different stimuli: KCl-enriched K-HS (60 mM) to induce depolarization-mediated contraction, the muscarinic agonist bethanechol (100 µM) to mimic the effect of ACh, the main neurotransmitter in the gastrointestinal tract, and 10 µM substance P, another contractile neurotransmitter in the colon. At the end of each experiment the strips were dried, weighed, and measured to normalize the contractile responses. The tension is given in milliNewtons (mN) normalized by cross-sectional area (CSA (mm^2^) = weight (g)/(specific density × length (mm), density 1 g/mm^3^)). All the contractility records were started at 03:00 p.m. local time, after reception and preparation of the sample (from 10:00 a.m. to 01:00 p.m. approximately).

### 4.5. Determination of Cytosolic Ca^2+^ Signals

After removal of the mucosa and submucosa layers by careful dissection the muscle layer was cut in small pieces and the muscle cells were isolated by enzymatic digestion with sequential incubation with papaine and collagenase following [44]. To perform real time estimation of changes in the cytosolic Ca^2+^ concentration ([Ca^2+^]_i_) we followed methods previously described [45,46]. Briefly, isolated cells were loaded with 4 µM fura 2-AM at room temperature for 25 min in a Na-HEPES solution (in mM: 10 HEPES, 140 NaCl, 4.7 KCl, 2 CaCl_2_, 2 MgCl_2_, and 10 D-glucose, pH 7.3) and then were placed in an experimental chamber made with a glass coverslip (0.17 mm thick) treated with poly-D-lysine to allow attachment of the cells after sedimentation. The chamber was mounted on the stage of an inverted microscope (Eclipse TE2000-S; Nikon, Barcelona, Spain) to excite the cells at 340 and 380 nm by a computer-controlled monochromator (Optoscan; Cairn Research, Faversham, UK) at 0.3 cycle/s, and the emitted fluorescence images were captured with a CCD camera (ORCAII-ER; Hamamatsu Photonics, Barcelona, Spain) and recorded using dedicated software (Metafluor; Molecular Devices, San Jose, CA, USA). The ratio of fluorescence at 340 nm to 380 nm (F340/F380) was calculated pixel by pixel as an index of [Ca^2+^]_i_. [Ca^2+^]_i_ signals were measured as delta increase in the ratio divided by previous resting ratio.

Once the experimental chamber is placed in the microscope stage, cells were kept under constant flow of Na-HEPES solution, and after a few minutes stimulation was achieved switching to solution containing the desired concentration of the stimulus (60 mM KCl) or delivering a small stream of the drug with a patch-clamp pipette close to the cell (caffeine 10 µM or ACh 1 µM) during a few seconds. The [Ca^2+^]_i_ response was evaluated as fold increase in [Ca^2+^]_i_ (∆R/R0) using the minute previous to the stimulation as resting value of the ratio of fluorescence (R0).

### 4.6. Statistics

Average results are given as average ± standard error of the mean (SEM). In addition to Pearson correlation, data were analysed using a segmented regression model following [47] using freely available software (SegReg, www.waterlog.info/segreg.htm). To evaluate differences between groups Student`s t test or U-Mann-Whitney test were used. Cosinor periodogram was performed using on line free software (Cosinor.exe, v 3.1, www.circadian.org) or cosinor.online. For display purposes, the 24 h recorded period of clock genes expression is represented twice.

## Figures and Tables

**Figure 1 ijms-21-00674-f001:**
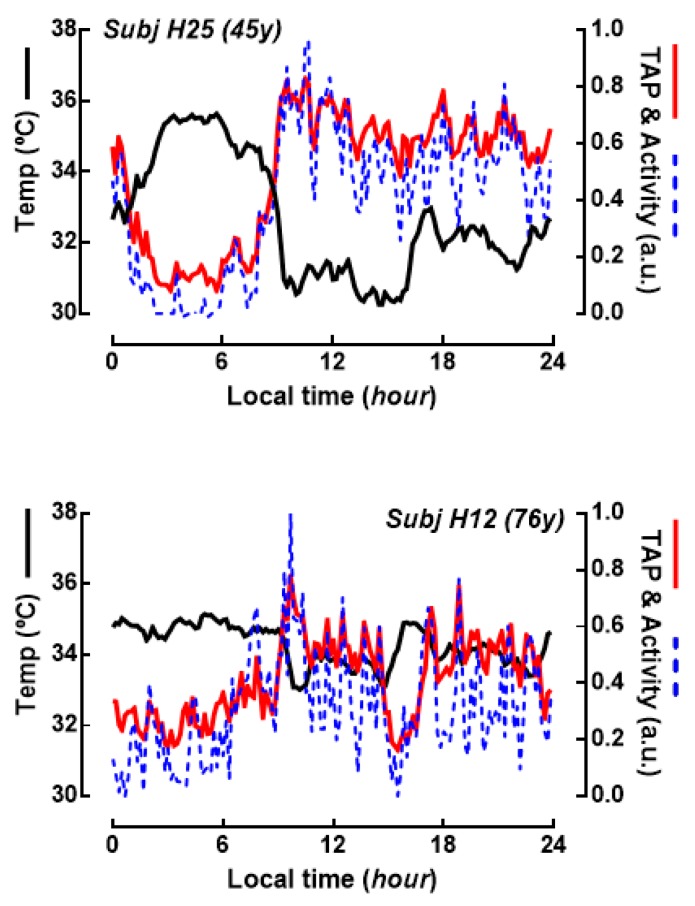
Circadian marker rhythms in human subjects. Traces represent the averaged values of skin wrist temperature and activity recorded ambulatorily during one week. The variable TAP (red line) is calculated from normalized records of temperature, activity and position (also recorded but not shown in this figure), and serves as a functional index of the circadian system in normal living conditions, oscillating from 0 (sleep, no activity) to 1 (wakeful, high activity). TAP and activity are shown as normalized records (arbitrary units) and temperature as raw data (°C). Top graph shows data from a 45 years old subject, and bottom graph from a 76 years old subject.

**Figure 2 ijms-21-00674-f002:**
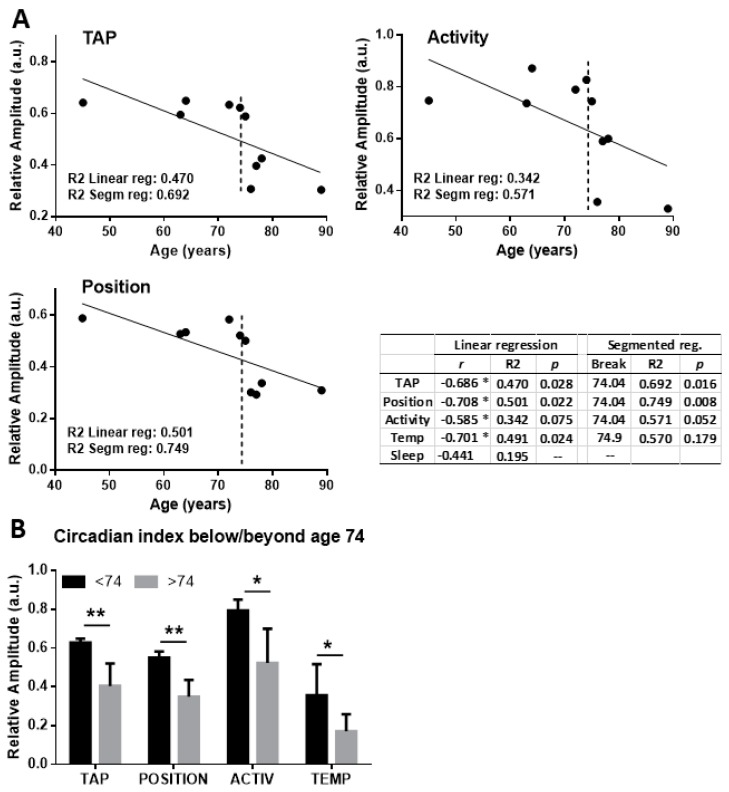
Aging is correlated with a decay in the marker circadian rhythms in humans. (**A**) Correlation between age and the Relative Amplitude (RA) of TAP, activity and position ambulatory rhythms in 10 individuals. The table shows the Pearson coefficient of correlation (r, where * *p* < 0.05) and the coefficient of determination (R2) and the level of significance (*p*) for both linear and segmented regression models. Column “*Break*” states the break point of the segmented model (indicated by a dashed line in the graphs). (**B**) Histograms showing a statistically significant difference in the RA index for the four main circadian variables before vs after 74 years age (* *p* < 0.05, ** *p* < 0.01, -- not significant). *n* = 10 individuals.

**Figure 3 ijms-21-00674-f003:**
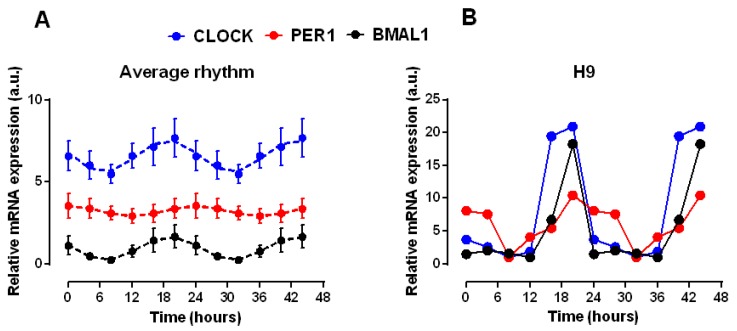
Circadian changes in the expression of *Bmal1*, *Clock* and *Per1* genes in smooth muscle explants from human colon. (**A**) Expression of *Clock*, *Per1* and *Bmal1* genes in colon muscle explants from 10 individuals was determined every 4 h during 24 h starting 8 a.m. the day following sample isolation. Data were averaged and simple cosinor analysis was performed (dashed line, cosine fitted function). For the sake of clarity *Clock* and *Bmal1* have been shifted upward (2.5 units) and downward (1.5 units) respectively to avoid overlap. (**B**) A representative record of genes expression. For display purpose, in both panels the 24 h recorded period is represented twice.

**Figure 4 ijms-21-00674-f004:**
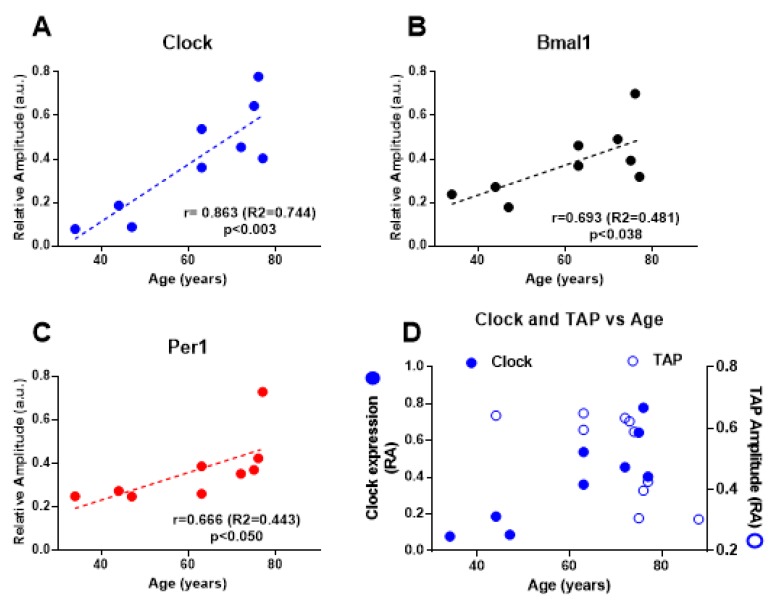
Aging increases the relative amplitude of the clock genes circadian rhythm in colon muscle. (**A**–**C**) Relative amplitude of the circadian rhythms of *Clock*, *Bmal1* and *Per1* expression was calculated for individuals of different ages. r: Pearson correlation coefficient; R2: determination coefficient; *p*: correlation significance. (**D**) Comparison of the age-related changes in the systemic circadian rhythm (TAP relative amplitude) and the colonic clock gene rhythm (*Clock* relative amplitude) in the same individuals.

**Figure 5 ijms-21-00674-f005:**
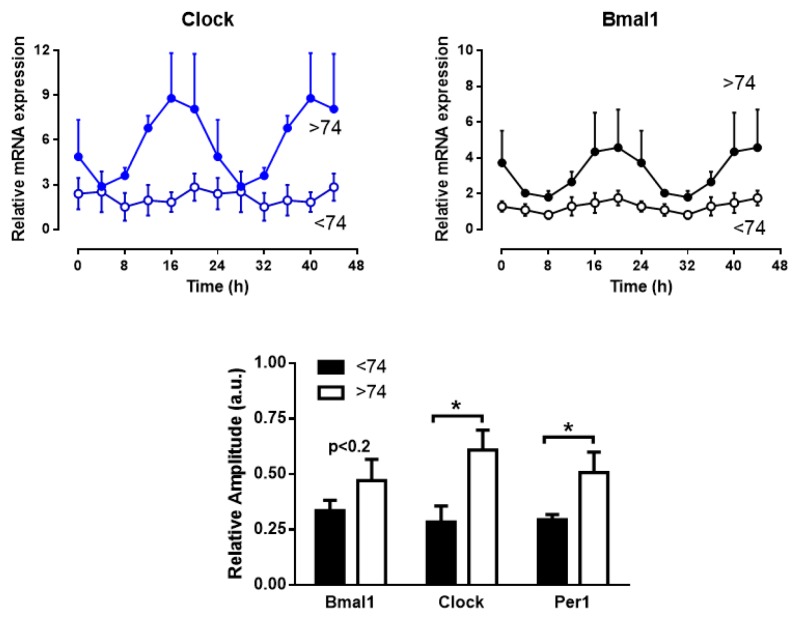
Clock genes rhythms in muscle colon are enhanced in subjects above 74 years old. The two top graphs show the averaged rhythms of *Clock* and *Bmal1* expression in colon muscle explants from donors aged below (*n* = 6) and above (*n* = 3) 74 years. The lower histogram depicts mean ± SEM of the relative amplitude of the rhythms for the two groups of age. * *p* < 0.05.

**Figure 6 ijms-21-00674-f006:**
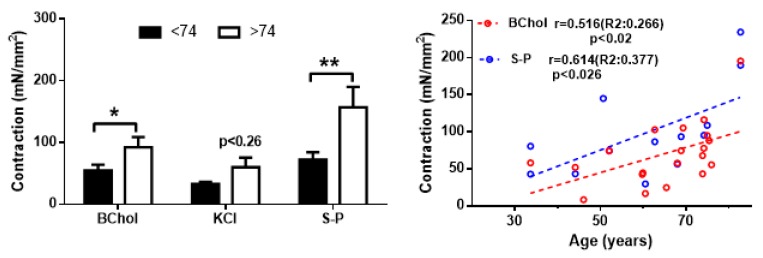
Age-related increase in the myogenic contraction of human colon muscle. Strips of circular colonic smooth muscle were challenged with 100 µM bethanechol (BChol), 60 mM KCl or 10 µM substance P (S-P). Left: Bars represent average ± SEM isometric contraction amplitude from 4–13 experiments from separate individuals. Right: correlation between age and contractility for bethanechol and substance P. * *p* < 0.05, ** *p* < 0.02.

**Figure 7 ijms-21-00674-f007:**
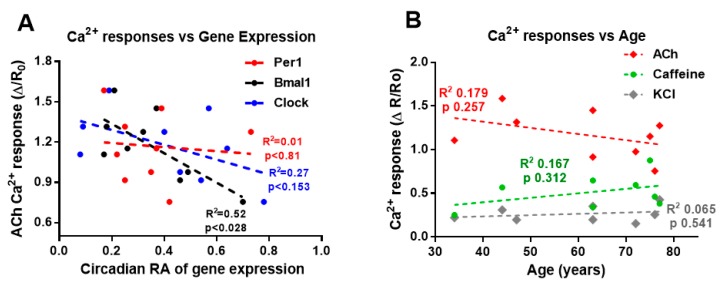
Ca^2^+ signals are not associated to aging nor to clock genes expression in human colon muscle cells. Smooth muscle cells isolated from the muscle layer of the colon were loaded with fura-2 to determine changes in intracellular Ca^2^+ concentration upon stimulation. (**A**) The average amplitude of the Ca^2^+ responses (fold increase) to ACh 1 µM is plotted against the relative amplitude of the clock genes rhythm for each subject. The R2 and p for correlation is shown. Only ACh vs *Bmal1* showed a significant negative relationship..*n* = 10–71 cells from 9 individuals. (**B**) Relationship between age and Ca^2+^ responses to ACh 1 µM, caffeine 10 µM and KCl 60 mM. Cells were averaged for each stimuli and individual. *n*= 9–26 cells for KCl, 5–22 cells for caffeine and 10–71 cells for ACh.

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
