# Peer review of "Age-Induced Differential Changes in the Central and Colonic Human Circadian Oscillators"

_ijms, 2020, doi:10.3390/ijms21020674_

Round 1

Reviewer 1 Report

This is an interesting study looking at circadian oscillators in peripheral tissues.  The fact that this has been demonstrated in humans is a big draw for the paper.  

The biggest weakness in the manuscript is in the description and presentation of the results.  This section needs to be significantly revised.  See suggestions below regarding the results section:

The narrative needs better structure.  The results need a clear description and the figures should assist in getting the message across; currently the figures are driving the section and this is impacting the flow. The TAP calculations needs to be better explained so that the reader can understand the significance of this variable.  Given suggestion 2, this will also help with the explanation of figure 1.   The axis in figure 1 need more explanation and the legend needs more description of the data- this is true of all the figures. There is an absence of means and SEM values in the description of the data.  If you are not going to show these on the graphs, you must give a feel of the SEM in the narrative. In figure 2, please add the r2 values on the graphs, this will help make it easier to interpret the results.   In the absence of SEM values in figure 3, it is difficult to assess if the rhythm shown is a true rhythm. For figure 6, the legend is not clear, are these age samples or young samples that are being analysed. 

In addition, I would also suggest that the authors bring out the fact that this discovery of the colon oscillator in humans is a novel finding- this will add to the impact go the study.  

Reviewer 2 Report

Camello-Almaraz et al. submitted to IJMS

Camello-Almaraz et al. investigated circadian variations of clock gene expression from ex vivo human colon, together with in vivo 24-h recording of actimetry and skin wrist temperature in the same subjects prior to colon sampling.

Major points

TAP used as phase-marker of the master clock

TAP is a variable, based on the integration of 3 rhythmic parameters, namely, skin wrist temperature, motor activity and body position. TAP has been developed to assess daily rhythmicity in human subjects exposed to usual living (i.e. uncontrolled) conditions.

However, it is not full why the authors consider that TAP is good marker of the master clock in the suprachiasmatic nuclei? What is the experimental evidence that TAP is a reliable phase-marker of the master clock? For instance, is it highly correlated with well-established hormonal markers of the SCN, such as plasma melatonin or cortisol rhythms?

The most obvious drawback of using TAP as a phase-marker of the SCN is that it does not allow dissociating between clock-controlled processes (eventually, but not necessarily depending on SCN, see below) and so-called masking processes (i.e., non-circadian, direct effects of light or social cues).

Moreover, the integrated parameters of TAP (i.e., skin wrist temperature, motor activity and body

Position) likely depend on many more circadian oscillators than the master clock itself. This is best exemplified by skin wrist temperature. Such rhythm may well depend on local clocks in the skin, among other peripheral clocks out of the SCN.

Incidentally, TAP, RA and CFI are mentioned in the main text (page 2) without explanation. To help the reader, please provide details (including spelling out) in Introduction and/or first sentences of Results section.  

Feeding schedule of studied subjects

Meal time is a potent synchronizer of peripheral clocks, including the colonic clock (Yoshida D. et al. 2015 Chronobiol Int). Thus, unusual feeding schedule may affect clock gene expression in human colon as well.

What was the feeding schedule of the studied subjects? What is the same between subject younger and older than 74 years-old?

Therefore, a potential effect of feeding schedule should be taken into account in the present results.

Line 42: the authors claim that the master clock in the SCN is entrained by meal time. However, the SCN is well-known to be insensitive to the entraining effects of meal time, as opposed to peripheral clocks (e.g. Damiola F. et al. 2000 Genes Dev; Stokkan KA et al. 2001 Science). Please reword accordingly.

Clock gene expression in human colonic muscle

According to the classical transcriptional-translational model of circadian oscillations, Per1 expression is supposed to peak in antiphase to Bmal1 and eventually to Clock. Here the acrophase of Bmal1 in human colonic muscle samples is phase-advanced only by 6 h or less, as compared to Per1 peak (Figure 3). Please comment this discrepancy in your article.

Because the main aim of the article is to investigate aging-related changes, it is not clear why mean patterns of clock gene expression are not shown in subjects younger than 74 years-old, as compared to older individuals (as done for TAP in Figure 2).

Expression levels of Clock, Per1 and Bmal1 mRNA were normalized to 18S levels. Did the authors check with cosinor analysis whether daily expression of 18S displays a significant rhythm, or not?

According to international nomenclature, gene names should be written in italics and lower case, not upper case as in the main text.

Other points:

Abstract: The first sentence is confusing (i.e. “Aging modifies multiple cellular and homeostatic systems, including biological rhythms”). Biological rhythms are usually considered as complementary of homeostatic systems (e.g. sleep which is regulated by both homeostatic and circadian processes). Please reword the sentence.

Line 52: “while there is”, instead of “while and there is”.

Legend of Figure 1: please specify that so-called temperature actually corresponds to skin wrist temperature.

Line 183: The human colon “presents”, not “present”.

Line 187: The circadian disruption discussed above “is”, not “are”.

Lines 189-191: The following sentence is quite confusing: “The role of chronodisruption in these changes is indicated by the preventive effects of melatonin in visceral smooth muscle”. The efficacy of melatonin treatment is certainly not a scientific argument to show/support that there is a chronodisruption and/or that chronodisruption plays any role. Please reword or delete the sentence.

Line 205: the target is “another”, not “other”.

Reviewer 3 Report

This paper provides new data on a poorly-characterized aspect of the circadian system of mammals: the peripheral oscillators that function in tissues outside the nervous system, their relationship to the central oscillator in the brain, and the effect of age on that relationship. In this case, the authors have looked at peripheral oscillators in the gut, and in a very unusual aspect of the work, they use human tissue instead of model organisms. They had access to a set of patients undergoing colon surgery and were able to obtain tissue samples to assess in vitro rhythmicity of clock genes. Patients were also assessed for central rhythmicity by temperature, activity and position. As predicted based on previous work by others, increasing age decreased the amplitude of central rhythms. Counterintuitively, increasing age increased the amplitude of clock gene expression in the tissue explants. The authors also found an increase in contractility of the tissue with age, which could not be accounted for by increases in calcium signalling.

This is very interesting work that makes a significant addition to our knowledge of peripheral oscillators. The work is on the whole well-conducted with appropriate methods and adequate data analysis. Some aspects of the paper require improvement, as follows:

Lines 28-29 and 151: “suggesting a loosening of the entrainment of the molecular clock of the colon by the central pacemaker”. This explanation of the increase in clock gene amplitude in the colon explants with age is not developed in the discussion, and it is not a convincing speculation. The clock gene rhythms were observed over greater than 24 hours in tissue removed from the influence of any SCN signals and cultured in vitro. The SCN entrains peripheral oscillators through neural and hormonal signals, but these are absent in vitro. All the samples, young and old, are free of entrainment signals from the SCN, so the effects of age must have some other cause. The lack of any adequate discussion on this point is a major deficiency in the paper. The authors may want to consider how the culture conditions might differentially affect old vs. young cells. For example, it is very probable that the procedure for putting cells into culture will reset the peripheral oscillators, probably by the addition of 20% FBS, since a serum shock is well-known to reset mammalian cells in culture. Do old cells respond more strongly to this serum shock and therefore start their in vitro oscillations at a higher amplitude? Alternatively, the authors may want to consider whether old cells differentially express some signaling components in other pathways that may enhance clock gene expression. In any case, the explanation must lie within the cells themselves and not in the entrainment pathway from the SCN.

Line 42: Meal times do not entrain the SCN. The food-entrainable oscillator acts independently of the SCN.

Line 52: delete the extra “and”

Figure 2: Labels A and B are needed.

Figure 4: There seem to be only 9 data points on A, B and C, and only 7 data points for Ratio in D, although there are 10 data points for TAP. Why?

Figure 4: Why is the Y axis in D a ratio of Clock/TAP amplitude instead of just Clock amplitude?

Line 121: “a wider series of experiments” is mentioned but there are no details about this. What does that mean? Were other patients sampled? If so, we need complete details in the methods.

Lines 125-126: “a slight positive correlation” - is this a linear model? It is confusing when the data are first described by a segmented model in Fig. 5. The plot for this correlation should be shown.

Line 127: There is no explanation of the method for electrical field stimulation and no data shown. This should be added.

Line 135: “three different agonists” - the details of treatment need to be added to methods, such as concentrations used and timing of the agonist addition to the cells.

Line 137: “no correlation” - show the data.

Line 137 and Figure 6: If Bmal and ACh response are negatively correlated, and Bmal and age are positively correlated, why are ACh response and age NOT correlated?

Line 140: “additional samples” - explain what these are.

Figure 6: Again, there seem to be only 9 data points although there were 10 patient samples.

Lines 189-190: “The role of chronodisruption in these changes is indicated by the preventive effects of melatonin” - this statement is too strong a conclusion. “Suggested” would be better than “indicated” - there are other alternative explanations for the effects of melatonin.

Lines 209-210: All subjects were, of course, scheduled for colectomy and must therefore have serious health problems, but there is no information about their health conditions that required this surgery. It is understandable that there are no “healthy” controls without colon disease in this human trial, but there is still the lingering worry that the patients’ underlying health issues may affect the results. Can the authors address this issue? Were older patients suffering from different health issues than younger patients and could this be a hidden variable that accounts for the correlation of age with clock gene expression?

Line 220: “in a bracelet, in a pendant” - this is confusing. Which sensors were in which device?

Lines 223-224: Please add more detail on the definition and calculation of RA and CFI.

Line 243: The description of this method leaves out any detail of the timing. Was the time of day of the measurements recorded? Or is it the expectation that the peripheral oscillators in these samples would be reset by the culturing protocol? If so, was the timing of the assay standardized for all samples? The worry is that differences in responses may be due to different phases of the internal oscillators in the tissue samples.

Lines 250-251: “two different stimuli” - isn’t it three different stimuli?

Line 273: The plus/minus sign has disappeared.

Round 2

Reviewer 1 Report

I am happy with responses and additional information provided by the authors to my previous comments.  The manuscript has improved by the additional information and clarity provided.  

Reviewer 2 Report

In their revised version, the authors adequately took into account most of my previous concerns.

Incidentally, gene names are supposed to be written in italics.

Below are listed additional typing errors :

Line 216 : did not show, instead of did not showed.

Line 246 : there is, instead of there si.